# Water-Activated Semiquinone Formation and Carboxylic Acid Dissociation in Melanin Revealed by Infrared Spectroscopy

**DOI:** 10.3390/polym13244403

**Published:** 2021-12-15

**Authors:** Zakhar V. Bedran, Sergey S. Zhukov, Pavel A. Abramov, Ilya O. Tyurenkov, Boris P. Gorshunov, A. Bernardus Mostert, Konstantin A. Motovilov

**Affiliations:** 1Center for Photonics and 2D Materials, Moscow Institute of Physics and Technology, Institute Lane 9, 141701 Dolgoprudny, Russia; bedran@phystech.edu (Z.V.B.); zs1978@mail.ru (S.S.Z.); abramovpa33@gmail.com (P.A.A.); turenkov1997@mail.ru (I.O.T.); bpgorshunov@gmail.com (B.P.G.); 2Department of Chemistry, Swansea University, Singleton Park, Swansea SA2 8PP, UK; a.b.mostert@swansea.ac.uk

**Keywords:** melanin, FTIR spectroscopy, water, comproportionation

## Abstract

Eumelanin is a widespread biomacromolecule pigment in the biosphere and has been widely investigated for numerous bioelectronics and energetic applications. Many of these applications depend on eumelanin’s ability to conduct proton current at various levels of hydration. The origin of this behavior is connected to a comproportionation reaction between oxidized and reduced monomer moieties and water. A hydration-dependent FTIR spectroscopic study on eumelanin is presented herein, which allows for the first time tracking the comproportionation reaction via the gradual increase of the overall aromaticity of melanin monomers in the course of hydration. We identified spectral features associated with the presence of specific “one and a half” C𝌁O bonds, typical for *o*-semiquinones. Signatures of semiquinone monomers with internal hydrogen bonds and that carboxylic groups, in contrast to semiquinones, begin to dissociate at the very beginning of melanin hydration were indicated. As such, we suggest a modification to the common hydration-dependent conductivity mechanism and propose that the conductivity at low hydration is dominated by carboxylic acid protons, whereas higher hydration levels manifest semiquinone protons.

## 1. Introduction

The melanins are one of the most diverse and widespread families of natural pigments in the biosphere [1,2,3]. For example, the pigments are found in plants, as well as animals. In lower vertebrates, the presence of melanin in the liver and spleen suggests that it has cytoprotective functions against cytotoxic species such as activated oxygen [4,5,6]. More famously though, the melanins are found throughout the human body, including the eyes [7], the ears [8], the substantia nigra of the brain stem [9], and the skin, where it acts as our main photo protectant [10]. 

There are two main forms of melanin, pheomelanin, which is red-yellow in color, and eumelanin, which is a brown-black pigment [11]. Notably, a third form, termed neuromelanin, is found in the brain stem and is a combination of pheomelanin and eumelanin [12]. Of the above categories, the most widespread melanin in the animal kingdom is eumelanin, and it is often considered the archetypal melanin [11]. Eumelanin is a polymeric material derived from two monomers, 5,6-dihydroxyindole (DHI) and 5,6-dihydroxyindole-2-carboxylic acid (DHICA) and their various redox states and tautomeric forms (Figure 1). 

Eumelanin, the focus of our work here and referred to as melanin hereafter, has received significant attention in the last decade owing to its unique physicochemical properties, which include: broad-band optical absorption [13,14], a persistent free radical signal [15,16,17], the ability to protect against harmful ionizing radiation [18], almost 100% nonradiative conversion of light energy [13], metal ion chelation [19,20], hydration-dependent conductivity [21,22,23,24], photoconductivity [21,25,26], and hydration-dependent electrical switching behavior [27,28,29]. Due to these aforementioned properties, melanin has been tested in numerous bioelectronic and bio-based applications, including transistor devices [30,31], biodegradable/biocompatible energetics including supercapacitors [32,33,34,35], and elsewhere [36,37,38,39]. Even though melanin is not a monolithic material and the ratio between DHI- and DHICA-based moieties in the polymer in the dry state depends on the particular synthetic procedure used/origin of material [40], it is clear that there is a significant effect on melanin’s electrical properties in the presence of water. With water present, a redox equilibrium (Figure 2) generates a semiquinone anion and a charged proton species, which leads to an enhanced electrical conductivity of the material [21,24,41,42]. Since conductivity is a crucial parameter for device applications, attempts have been made to modify it via non-hydration means such as with metal ion chelation [31] and sulfonation [43]. Vibrational spectroscopy, including Raman scattering and FTIR techniques, except inelastic neutron scattering (INS), has been used to characterize melanin materials numerous times [4,15,44,45,46,47,48,49,50,51,52,53,54,55,56,57,58]. Particular efforts were aimed at clarifying the origination of the spectral peculiarities of melanin from definite monomer units [52,59], dimers [49,60], and tetramers [49,58]. Only a few studies were performed with precise attention towards the level of hydration of the material and the corresponding consequences for the spectral features [44,50,61]. The first attempt to use the inelastic neutron scattering (INS) technique for probing the water morphology in hydrated melanin was recently performed [62]. It demonstrated the features unique among other INS-based studies of bioorganic materials and included observation of the bands attributed to the aqueous proton cations hydronium H_3_O^+^ and Zundel H_5_O_2_^+^. However, the authors could not track the complex dynamics of these cations’ concentrations in the course of hydration due to the insufficient frequency resolution of the data. Furthermore, another limitation is that the INS technique demands low temperatures of 20 K and below. This condition brings obvious difficulties to the extrapolation of the obtained result to room temperature, especially for systems with thermodynamically labile chemical equilibria including hydrated melanin.

We could not find any study focusing on the spectral fingerprints of the gradual deprotonation of carboxyl groups, semiquinone hydroxyls, and other possible sources of mobile protons in melanin in the course of hydration.

To address some of the shortcomings in the literature regarding the nature and sources of the protonic charges, we report here a Fourier transform infrared (FTIR) study of synthetic melanin films. The films were probed under thoroughly controlled levels of hydration at room temperature (25 °C) to observe the behavior of the melanin and the water. FTIR is an excellent tool to investigate the potential formation of different hydrated protonic charges since each species will have a different vibrational signature. 

Unlike previous vibrational spectroscopy studies, the current work is aimed at revealing the signatures of carboxylic acid deprotonation and comproportionation reaction leading to the formation of mobile proton species (Figure 2). We go on to juxtapose our observations with already published data on hydration-dependent AC and DC conductivity [15,22,23,24,42,63], muon spin resonance (muSR) [21], electron paramagnetic resonance (EPR) [63,64,65], and inelastic neutron scattering [62].

## 2. Materials and Methods

### 2.1. Melanin Synthesis

Melanin was synthesized following a standard literature procedure [19] utilizing as the initial starting material D,L-dopa (Sigma-Aldrich, Burlington, MA, USA). D,L-dopa was dissolved in deionized water, subsequently adjusted to pH 8 using NH_3_ (28%). Air was then bubbled through the solution while being stirred for 3 d. During the 3 d synthesis, the pH would naturally decrease due to the evaporation of NH_3_, which would necessitate the need to add ammonia periodically to bring the pH back to 8. By keeping the pH at a maximum of 8 and letting it decrease naturally ensured that ring fission of the indolequinone moieties were kept to a minimum, and hence, the synthesized melanin was a biomimetic material [66]. The solution was then brought to pH 2 using HCl (32%) to precipitate the pigment. The solution was then filtered and washed multiple times with deionized water and dried. We made 4 batches and then homogenized them to make a representative material.

### 2.2. IR-Measurements

Transmission spectra in the mid-infrared region (MIR) were obtained by utilizing a Bruker Vertex V80 FTIR spectrometer (Bruker, Billerica, MA, USA) with a Hyperion 2000 microscope console, equipped with an IR objective of ×36 (NA 0.5, working distance 10 mm) and an 80 micrometer aperture.

Humidity control was achieved using a custom hygrostatic system, as shown in Figure 3A,B. The so-called hygrocell (1) containing the eumelanin thin film was supplied with moisture by a saturated salt solution air from the container (2). A small fan was installed inside the salt solution container to create a steady moisture distribution throughout the volume. A modified APS 300 pump (Tetra Werke, Melle, Germany) (3) was used to circulate the moisturized air through the system. A humidity sensor was located inside an empty container (4) equipped with a fan. The saturated salt solutions used to achieve humidity with their corresponding relative humidities (RH) can be seen in Table 1. The salts were obtained from Sigma-Alrich, Burlington, MA, USA (LiCl, KCl) and Rushim, Moscow, Russia (MgCl_2_, Na_2_Cr_2_O_7_, K_2_CO_3_, NaCl) and used as received.

The minimal value of the relative humidity was achieved by pumping the hygrocell to a pressure of 5 × 10^−5^ mbar. The samples were incubated at this pressure for 1.5 h. The relative humidity of 1.4 % was obtained by venting the optical hygrocell for 1 h with gaseous nitrogen containing the corresponding level of humidity. We estimated the moisture content in nitrogen via mid-infrared transmission measurement with a good resolution (0.2 cm^−1^) in a Bruker Vertex V80 sample compartment (Bruker, Billerica, MA, USA). The water content was calculated by applying the line spectral strength analysis using the HITRAN [67] open spectroscopic database and the load_hitran MATLAB function [68].

For spraying, we prepared the solution in 20% aqueous ammonia with synthetic eumelanin with a concentration of 45 mg per 1 mL. The solution was stirred for 1 h and then ultrasonicated for 1 h as well. Then, the solution was sprayed via an airbrush from a distance of 10–15 cm onto a CaF_2_ substrate previously heated at 50 °C and cleaned with isopropyl alcohol. We utilized a nozzle with a diameter of 0.5 mm. The air pressure in the airbrush was 1 atm. The thickness of the sample films was estimated utilizing a SOLVER NEXT Scanning Probe Microscope (NT MDT, Zelenograd, Russia) and was found to be around 1.5 µm. Since the film was inhomogeneous, we collected spectra from 6 different positions (Appendix A).

To achieve equilibrium between our sample and the moisturized atmosphere, we incubated it at the desired humidity level until the humidity level sensor started to show a constant signal for at least 20 min. Then, we made two test MIR spectra measurements with a time interval of 10 min. If two sequential test spectra gave the same signal, we concluded that the equilibrium had been achieved. Otherwise, we waited for 10 min and measured the spectra again. 

To test the system for the reversibility of the spectral response in the MIR range depending on humidity, we increased it from 0% to 84% and then back to 0%, utilizing the saturated salt solutions indicated in Table 1.

### 2.3. Melanin Characterization

The samples were confirmed as melanin via UV-Vis absorbance spectroscopy, electron paramagnetic resonance (EPR), and X-ray photoelectron spectroscopy (XPS).

For the UV-Vis absorbance spectroscopy, a small amount of melanin powder was dissolved in a pH 8 deionized water solution (adjusted with NH_3_). The spectra were then obtained utilizing a Perkin Elmer PDA UV/Vis Lambda 265 (Perkin Elmer, Waltham, MA, USA), using the wavelength monitoring functionality, obtaining a wavelength range from 350 nm to 900 nm at 1 nm wavelength intervals. Example data can be seen in Figure 4A, and the featureless, exponential decay spectra were as expected for melanin [69].

For the EPR measurement, a powder sample of the melanin was measured using a Bruker EMX Micro X CW-EPR spectrometer (Bruker, Billerica, MA, USA) with an E4104 X-band cavity at a microwave power of 0.87 mW and at room temperature. The spectra were taken at a modulation frequency of 100 kHz and modulation amplitude of 1 Gauss with a scan width 60 Gauss. The apparent isotropic g factor was calibrated against a DPPH standard and found to be 2.0036, which is as reported elsewhere for solid melanin synthesized in the same manner [43]. An example spectrum can be seen in Figure 4B.

An elemental analysis for melanin can be reliably probed via XPS. For the XPS, a wide-scan survey spectrum was performed on pressed pellets of the powder utilizing a Kratos Axis Supra (Kratos Analytical Ltd, Manchester, UK) using 225 W AlKα X-rays with an emission current of 15 mA and equipped with a quartz crystal monochromator with a 500 mm Rowland circle. Spectra were collected with a pass energy of 40 eV, with the hybrid lens setting, a 0.1 eV step size, and a 1 s dwell time for electron counting at each step. To eliminate differential charging [70], the charge neutralizer integral to the Kratos spectrometer was used as an electron source. 

### 2.4. Data Processing

We divided the analysis of the infrared spectra into two parts. The first one was devoted to the deconvolution of the measured spectra into an appropriate set of Gaussian lines. This part aimed to obtain the dependence of the parameters of the Gaussian lines from the humidity level. The second part of the analysis aimed to refine the amount and central wavenumbers of the Gaussian lines by varying the second derivative calculation parameter. 

Part 1. RH dependence:

The measured infrared spectrum was deconvoluted into an appropriate set of Gaussian and Lorentzian spectral lines following the procedure below using the Fityk data processing software. At the preprocessing stage, we subtracted the baseline from all absorption spectra caused by scattering and reflection. As one of the most powerful tools for the detection of overlapping peaks, we used the second-derivative peak-finding procedure [71]. The central concept of this approach utilizes the fact that the local extremum positions of the Gaussian lines are equal both in the transmittance (or absorbance) and in the second derivative (SD) of the transmittance (or absorbance), which is more sensitive to the position of the peak. The procedure involves collecting the peak positions in the SD graph for the spectra, measured at all six spatial positions on the sample and all humidity levels. We then used the SD peak positions as an initial guess for the peak positions in the fitting procedure for the first measured position on the film and for RH = 0% for the Gaussian line basis set. We estimated the corresponding Gaussian peak widths and heights manually. We then utilized a Levenberg–Marquardt least squares difference minimization algorithm (L-M LSDM) to obtain a refined result for all parameters. We found that a set of Gaussian lines fit well with the experimental data except for the mode with the 1724 cm^−1^ peak position. Due to its wide slope, we decided to use the Lorentzian line shape instead for this particular line. The derived parameters were used as an initial guess to process the data obtained for higher humidities and the other measured spatial positions on the film. Since we did not observe reliable frequency shifts during hydration in the second-derivative graph (Appendix A), we fixed the frequency positions for all modes. Moreover, we optimized the parameters of 2866 cm^−1^, 2929 cm^−1^, and 2968 cm^−1^ only for RH 0% and fixed all the parameters for the remaining RH values to increase the fitting accuracy. Such an assumption came from trial fits that showed the low dependence of those parameters from the RH and the strong interplay between these parameters and the low line strengths of these excitations. Finally, we obtained the dependence of the parameters of the lines from the RH value for all measured positions on the film.

Part 2. Central wavenumber refinement:

Deconvolution of the spectra into a set of lines usually requires a balance between the fitting precision and the number of lines in a model. Indeed, for example, the set of Gaussian lines forms a complete system of functions, which means that we can construct a model from it that fits the experimental spectra with any predetermined accuracy. However, in the real situation, we will then face the optimization problem of all model parameters.

The second derivative of the measured IR spectra could give us complete information about the number of peaks and their positions. However, in the real situation, the direct calculation of the SD with some finite-difference algorithm will be strongly affected by the noise. There are several ways to overcome this issue. We chose the total variation differentiation algorithm [71]. It is based on a Tikhonov variation paradigm to solve an inverse problem on noisy data. The main adjusted parameter in this algorithm is the α (regularization parameter), which controls the balance between the irregularity term and the data fidelity term in the functional to minimize [71]. 

In the previous part, we chose the α in such a way so as to only see peaks with a width of more than 15 cm^−1^, indicating α = 5. We found that this enabled us to fit our model well with the data with no mutual dependencies between fitting parameters. However, our further aim was to obtain the number and frequencies of peaks in our spectra. To do that, we varied α in the range from 400 to 10^−6^. We found that starting from α = 10^−2^, the SD graph was strongly affected by the errors caused by water vapor and CO_2_ between the microscope and sample (Appendix A). This required us to reduce the range of α’s variation down to 400−0.1.

In the case of large α = 400−25, we detected several broad and weak peaks. In the case of low α = 1−0.1, we recognized the lines consisting of several peaks (Appendix A). 

## 3. Results and Discussion

The characterization results of our sample utilizing UV-Vis, EPR, and XPS can be seen in Figure 4 and Table 2. The UV-Vis data in Figure 4A show a featureless, exponential decay spectrum, as expected for melanin [69]. The EPR spectrum shown in Figure 4B has an apparent isotropic g factor, calibrated against a DPPH standard, of 2.0036, which is as reported elsewhere for solid melanin synthesized in the same manner [43]. As shown in Table 2, the atomic composition (atomic concentration in at% and atomic ratio) of the sample can be seen. Melanin is an oligomeric mixture primarily composed of the DHI and DHICA moieties and their assorted oxidative states [11,72,73]. Consequently, it is expected that melanin samples will have an overall atomic ratio profile that falls in between those expected for DHI and DHICA (see Table 2, “Expected”). The XPS data indicate that the sample is compatible with that of synthetic melanin (Table 2; see “Sample”). The sample is closer to the ideal DHI values, indicating the dominant ion of the DHI moieties. This was expected for the synthesis procedure employed here [66] since it is known that there is a loss of COOH units, as well as the formation of a small amount of smaller ring units due to degradation [74]. We also note that elemental surface scans of melanins are representative of the bulk, as previously demonstrated [75]. Overall, the characterizations above show that our sample is a synthetic sample of eumelanin.

Since the peaks on the curves of the second derivative of transmission had no dependence on the moisture content (Appendix A), we concluded that the absorption peak frequencies remained fixed as well. The peak positions and their assignments are listed in Table 3.

Figure 5 shows the FTIR absorbance of the melanin film equilibrated with the correspondingly moisturized atmosphere in the wavelength range of 1000–4000 cm^−1^. Shown also are the deconvoluted spectra following the second-derivative line-detection procedure as described in the Materials and Methods.

We did not find appropriate assignments for modes with central frequencies of 1882 cm^−1^, 2092 cm^−1^, and 2224 cm^−1^. However, in [56], one may observe several broad and weak peaks in the region of 1800–2300 cm^−1^. More important is that the number and position of those peaks enormously vary from one sample to another. That is why we concluded that these peaks are due to some residuals of the synthesis.

In the further analysis, we wanted to derive the concentration of the infrared radiation absorber following the Beer–Lambert–Bouguer law, which states that the absorption coefficient at a particular frequency is a product of the optical path length, absorber concentration, and molar extinction coefficient at that frequency. However, the peaks from different absorption species often overlap and thus make the direct calculations of the concentration impossible. To overcome this problem, one needs to deconvolute the spectra into an appropriate set of well-known line shapes and then analyze them separately. To reduce the noise effect on the calculations, the integral of the absorption coefficient over the mode is usually used instead of the absorption coefficient at a particular frequency itself. The derived value is called the line strength (*L*) and is measured in cm^−1^. Thus, the corrected form of the Beer–Lambert–Bouguer law for the line strength will be as follows: *L = x × c × ε*, where *L* is the line strength, *x* is the optical path, *c* is the sought concentration of the absorption species, and *ε* is the integral molar extinction coefficient. The determination of *ε* is possible for simple chemical objects. However, in the case of our material, we do not know the *ε* of the different absorption modes. Therefore, direct estimation of the absolute values of the concentrations is impossible. However, we analyzed the relative change of the line strength by normalizing it (i.e., the peak area) to the average line strength value obtained across all levels of relative humidity. As a result, we obtained the value directly proportional to the relative change in the concentration of absorbing species. The uncertainty bars were estimated from the L-M LSDM fitting algorithm. The uncertainty in the raw data due to noise was neglected as we found it to be much less than the uncertainty due to the fitting (0.2% for noise and >1% for fitting). The complete set of hydration dependencies of the line strengths is presented in Figure 6. In Appendix A, the same dependencies are normalized to 84% RH and 0% RH correspondingly. 

Before the discussion, we would like to make several remarks. The composition of melanin includes a relatively large set of various functional groups, even without considering the water structures, having overlapping spectra in many cases. In this regard, certain caution is required with the final interpretation of the result and knowledge obtained by other methods about the changes in the properties of the material accompanying the hydration level increase. Furthermore, due to obvious methodological limitations, we could not track the changes in the intensity of all the lines detected from the spectra of the second derivative of optical transmission (Appendix A), depending on the humidity. Only those bands obtained via the deconvolution of the direct absorbance spectra could be effectively analyzed in terms of the dependence on the hydration level (first column in Table 3).

We first should mention that the line strength dependency level varies from one measured position to another. For some modes (for example, 1033 cm^−1^, 1111 cm^−1^, 2675 cm^−1^), those changes are larger than 100%. We can explain this by stating that our film is inhomogeneous, not only in thickness, but also in chemical composition. However, if we normalize each graph to its value at 84% (Appendix A), we could see that the shapes of the dependencies are consistent between measured positions on the film. This means that local variations of the chemical composition of our film do not affect the chemical processes occurring in our sample on a qualitative level.

We turned to inspect the relative changes in the lines’ strength, which can be evaluated to see whether humidity affects the concentration of the corresponding group or not. We classified the data according to the following criteria. For example, from the presence of hysteresis on the curves of the dependence of the line strength on humidity (Figure 6 and Appendix A), it can be seen that the greatest hysteresis is observed for the bands of 1363 cm^−1^, 1404 cm^−1^, 1454 cm^−1^, 1514 cm^−1^, 2890 cm^−1^, 3444 cm^−1^, and 3588 cm^−1^. Only for the bands of 3444 cm^−1^ and 3588 cm^−1^ do we see a good coincidence of points at the beginning and end of the moistening and drying cycle. Since the concentration of the components responsible for the last two absorption bands increases many times in the material upon moistening, it is logical to attribute them to the vibrations of water molecules or water cations. The stretching vibrations of the N–H and organic O–H groups can also be present in this range. However, such a drastic increase of their concentration during hydration is unreasonable since simple double-bond water attachment is impossible under our conditions. Nitrogen already has its hydrogen in the oxidized quinone tautomer and quinol (Figure 1I,VIII). The appearance of two water-related bands of O–H stretching vibration can be explained by the co-existence of hydrogen-bonded and free hydroxyls in water and by shifting of the corresponding vibration in aqueous cations such as hydronium H_3_O^+^ or Zundel H_5_O_2_^+^.

Turning to the bands with high hysteresis but different states at the start and the end of the hydration–drying cycle, the first thing that can be seen is the similarity of the dependencies for the bands of 2890 cm^−1^ and 1404 cm^−1^. Each curve increases at first and then goes down. The dependence for the 1454 cm^−1^ band behaves similarly, but not as pronounced, which demonstrates much less growth in the initial section. Earlier, Roldán et al. [56] attributed the 1404 cm^−1^ and 1454 cm^−1^ bands to the δ (OH) + ν_ring_ and ν (ring) + ν (CN) + δ (NH) + δ (OH) combined vibrations correspondingly (see Table 3). Centeno and Shamir [48] attributed the 1404 cm^−1^ band to pyrrole ring stretching and 1454 cm^−1^ to C=C aromatic ring vibration, whereas Bridelli et al. [61] assigned the 1404 cm^−1^ band to carboxylate ion symmetrical stretching, but did not observe the 1454 cm^−1^ band. We tend to agree with Bridelli et al. that water should induce active deprotonation of –COOH groups in DHICA, leading to the accumulation of –COO^−^ anions. As discussed below, the proton dissociation from carboxylic acid is manifested via other bands. However, the absence of an asymmetric stretch of the carboxylate ion is a mystery. This problem was not discussed by Bridelli et al. Usually, such a line is stronger than the symmetrical stretch [78] and is manifested in the 1550–1610 cm^−1^ region. In our case, we also did not have signatures of strong absorption in this range. The second-derivative spectra did not give any band signatures in this range at reasonable values of parameter α (Appendix A).

In the following, we offer an alternative interpretation of the 1404 cm^−1^ and 1454 cm^−1^ bands and their satellite manifestations in the second derivative of transmission at 1380 cm^−1^ and 1468 cm^−1^, respectively. According to multiple studies of *o*-semiquinone-based complexes, the broadband range of 1420–1470 cm^−1^ is due to stretching of the semiquinone “one and a half” C𝌁O bond [79,80,81]. Additional shifting of the corresponding vibrations to lower frequencies (1404 cm^−1^ and, probably, 1380 cm^−1^) can be caused by the formation of intramolecular hydrogen bonds between oxygen radicals and hydroxyl groups (Figure 7IVc,VIc). This is an essential consideration since the importance of intramolecular H-bonds for the photoconductivity of melanin-like systems was recently raised in the work of Grieco et al. [82]. Correspondingly, we suggest that a wide 2890 cm^−1^ band contains O–H stretching vibrations of semiquinones with intramolecular hydrogen bonds. We should also mention here that in accordance with the study (Scheme 2 in [83]), at room temperatures, simple *ortho*-benzosemiquinones should relax to a single more stable form of intramolecularly H-bonded semiquinone. This implies that some form, either IVc or Vlc, is most likely redundant. Suppose we accept the semiquinone-based interpretation of 1404 cm^−1^, 1454 cm^−1^, 1468 cm^−1^, and 2890 cm^−1^ bands as the main one. In that case, why is there such a significant discrepancy between the increase of the concentration of semiquinone with and without the intramolecular hydrogen bond upon hydration? We believe the following explanation will suffice. Inside the melanin nuclei, semiquinones predominantly form intermolecular hydrogen bonds due to a large number of neighbors. At the interface, the probability of the formation of intramolecular bonds is higher since the number of neighbors is less for each monomer. With the initial addition of water, an active formation of semiquinones occurs mainly on the surface layer. This is supported by the rise of the curve for the bands at 1404 cm^−1^ and 2890 cm^−1^ in the range 0–33% RH. A further increase of the water concentration (RH values above 33%) should continue to catalyze the comproportionation reaction. However, this also leads to partial deprotonation of the neutral semiquinone, both with intramolecular and intermolecular hydrogen bonds, and to the formation of the semiquinone anion (Figure 1VII). Depletion of the protonated semiquinone forms decreases the line strength of the corresponding bands (1404 cm^−1^, 1454 cm^−1^, 1468 cm^−1^, 2890 cm^−1^) for RH values above 33%.

Additional, clear signatures of the increasing aromaticity of melanin at higher hydration levels come from the bands 1363 cm^−1^, 1514 cm^−1^, 3070 cm^−1^, and 3247 cm^−1^. In the 1363 cm^−1^ band, there are several contributions. However, the main one is δ(OH) coming from fully reduced indole systems (Figure 1VIII). A two-step decrease of this line strength looks as an inverted curve of melanin conductivity dependence on hydration level [21,23]. It corresponds to reducing indolic monomers concentration due to the comproportionation reaction.

As suggested earlier [56], we regard the 1514 cm^−1^ band as a feature that originated from the aromatic ring system.

Turning to 3070 cm^−1^, it should be noted that in the structure of melanin monomers, the amount of hydrogen atoms bonded to carbon atoms remains independent of the course of the comproportionation reaction (Figure 2). There are always three of them per intact monomer. The classical stretching vibrations for such C–H bonds in alkenes are slightly lower in frequency than the frequency of the 3070 cm^−1^ band we observed, for which we managed to obtain the dependence on humidity. In cycloalkenes, only extremely strained cyclopropene can give C–H stretching at a frequency of 3075 cm^−1^ [78]. On the other hand, in conjugated aromatics, the frequencies are somewhat higher, reaching 3080 cm^−1^. In this regard, we are inclined to believe that an increase in the strength of the 3075 cm^−1^ line with increasing humidity corresponds to an increase in the number of hydrogens that make up the aromatic carbon skeleton, as it should be per the comproportionation. As a side note, the band 3018 cm^−1^ visible in the second derivative of absorption is very close to the C–H stretching in cyclohexene (3020 cm^−1^), which is closer to a quinone configuration. We infer that it signifies C–H stretching in oxidized monomers.

In its turn, the band 3247 cm^−1^ can be associated with N–H and O–H stretchings in water and organic hydroxyls. However, it does not increase as steeply as the water-associated bands of 3588 cm^−1^ and 3444 cm^−1^ do. Moreover, it has no hysteresis and demonstrates a significant line strength even in the dry state. Therefore, we should exclude water as a possible source of the 3247 cm^−1^ band. It is also unlikely that the band is caused by hydroxyl O–H stretching since we did not observe any C–O stretching band with a similar pattern. Furthermore, all other bands associated with comproportionation reaction (Figure 2) leading to hydroxyls synthesis and dissociation demonstrate strong hysteresis. Another potential option is a recently proposed radical moiety within melanin, a nitrogen defect, which would yield a protonated amine (N–H_2_^+^) [84,85]. At this point in time, there is no further information on these structures except as suggested by electron paramagnetic resonance. We also inferred that if having to choose between N–H and N–H_2_^+^, the protonated unit would likely be a significantly smaller population vis-à-vis the amine, and as such, we are inclined to ascribe the signal to the amine. Overall, these considerations favor the N–H stretching origin of the line. Its dependence on hydration also corresponds to the comproportionation since the reaction (Figure 2) leads to the increase of hydrogen connected with conjugated nitrogen atoms in indolic nuclei.

In our opinion, the reason for the significant discrepancy between the start and end points of the moistening-drying cycle for all lines with large hysteresis is common. Skeletal rearrangements in melanin caused by comproportionation are accompanied by changes in the structure associated with the repulsion of like charges. Reverse relaxation is difficult due to the high viscosity of the polymer and implied long relaxation time. This property of melanin, responsible for the difficulty of working with it, is well known from transport measurements and experiments with electron paramagnetic resonance (EPR). In reasonable times, the complete regeneration of the structure is possible only through heating (in a vacuum). However, the simple proton migration related to the band 3247 cm^−1^ is a fast process and does not give hysteresis nor carbon-bonded protons in the case of the 3070 cm^−1^ band.

From our point of view, the absence of hysteresis in line 1614 cm^−1^ is due to the fact that the main contribution to it is associated with the oxidized forms of monomers. Their number can be more orders of magnitude than reduced and semi-reduced monomers [86]. Against this background, the synthesis of semiquinone and the depletion of oxidized monomers (I-III, Figure 1) in the comproportionation reaction remains unnoticeable. Similar reasoning can be applied, in general, to the bands 1033 cm^−1^, 1111 cm^−1^, 1214 cm^−1^, and 2675 cm^−1^, where the contribution from oxidized monomers is overwhelming. However, one comment on the 1214 cm^−1^ band should be given, and here, we come to the bands related to DHICA-based monomers.

There are two bands among those in Figure 6 showing a very fast decrease of the line strength in the course of hydration: 1724 cm^−1^ and 3354 cm^−1^. Both of them have low strength. We attributed these bands to carbonyl stretching C=O vibration and to O–H stretching within the carboxyl group. It is essential to underline that in both cases, the frequencies correspond to the carboxyl having hydrogen bonds [78]. This interpretation opposes that given earlier in [48,56], where the band 3354 cm^−1^ was attributed to ν(NH) stretching. The 1214 cm^−1^ band includes the C–OH stretching vibration of carboxyl. However, this contribution is manifested well only on the left side of the curve, where the drop is seen when the humidity changes from 0–11% RH. Due to the fast decrease of the -COOH groups’ concentration in the material at higher hydration levels, this contribution becomes negligible.

The overall hydration-dependent behavior of these three bands, 1214 cm^−1^, 1724 cm^−1^, and 3354 cm^−1^, suggests that DHICA-based monomers are the primary source of the protons in melanin at RH values below 33%. The semiquinone source of protons becomes important above 43% RH (2890 cm^−1^ band). This two-step proton doping of the material largely explains the classical behavior of the dependence of the conductivity in the material [21,23]. This mechanism can also be reconciled to previous muSR data, which showed that the free radical content tracked the conductivity. However, contrary to the initial hypothesis, we believe that the data suggest that the initial free radical increase is due to the formation of the protonated semiquinone per comproportionation, but that deprotonation of these radicals and the additional formation of anionic semiquinones occur only at higher values of RH. We also note that in such a two-step mechanism, the concentration of dissociated protons increases, but also the concentration of localization sites, anions of carboxyl groups, and anions of semiquinone, between which charge transfer may occur.

Other important conclusions can be made about the peculiarities of hysteresis in each particular band strength in Figure 6. We see, for example, that in the case of the 1514 cm^−1^, 3444 cm^−1^, and 3588 cm^−1^ bands, the line strength is higher during the drying process. In the other cases, we usually see an inverted image—the blue curve is under the green one. In general, when a proton is only migrating within the monomer during tautomerization, hysteresis is almost absent, but when a proton dissociates (from –COOH or from semiquinone), the reverse reaction is inhibited. On the other hand, the difference between the cyclic and open semiquinone configuration bands and the large hysteresis of the 1454 cm^−1^ band speaks in favor of a complex migration. By this, we do not mean to infer a diffusion of indole units since these would be locally confined, but instead an effective change in the semiquinone concentration gradient within a melanin particle during hydration. The potential mechanism we turn to below. We speculate that the act of hydration pushes a significant part of the semiquinone concentration from the depth of a melanin particle to the particle surface (Figure 8). Once congregated there, a hysteresis effect manifests when the drying process is applied since the semiquinones are unable or restricted to restoring to an open configuration, i.e., more time is required for the semiquinone concentration to return to the initial state. A potential reason for this hysteresis is that pi–pi stacking of the aromatic system is disrupted by the semiquinone formation; hence, reforming the initial morphological state is difficult, similar to speculations elsewhere [64].

The one issue for interpretation is the absence of a semiquinone anion fingerprint band. Partially, it can be attributed to the 1514 cm^−1^ band, which is regarded as a feature of growing aromaticity, but this reasoning faces an apparent physical challenge. Usually, anions soften the corresponding frequency of the C–O vibration. In our case, the situation is the opposite. However, it is most likely that the semiquinone anion contribution is masked among those bands manifested in the second derivative with the shifted frequency position, for example the 1380 cm^−1^ band.

It should also be noted that we did not observe completely new absorption bands in the spectra of dry and wet melanin. Almost all of them have been previously described and interpreted in some way. The importance of our study is due to the fact that we carefully investigated the behavior of these lines in light of a systematic hydration change. Considering the fundamentally new data on the structure of melanin, on the structure of water at the interface and on the course of the comproportionation reaction as obtained in recent years, we link many spectral features to an amended model of the comproportionation reaction.

Turning to the issue of the underlying mechanism of the semiquinone concentration changes within a melanin particle, technically, the apparent migration of the semiquinones within the chain of oxidized monomers can be regarded as a defect migration in the form of the sequence of redox reactions resulting in the transfer of one electron and one proton (Figure 9). There are many options and details for the particular mechanism. However, we tend to speculate that this is a concerted proton–electron transfer (CPET) [87].

To understand the case for a CPET mechanism, we discuss the problem of charge conductivity for melanin. Previous results demonstrated that proton mobility should have no dependence on water concentration [21]. Indirectly, this speaks in favor of the fact that the network of hydrogen bonds formed by water does not affect the speed of proton movement. Our recent study of the structure of water in melanin using inelastic neutron scattering also suggests that water is in a disordered state in this material [62] and, thus, supports the earlier data. Furthermore, in 2015, it was demonstrated that from the viewpoint of DC conductivity, the kinetic isotope effect in melanin is below one, since H_2_O/D_2_O exchange during hydration leads to the slight increase of conductivity [63]. The latter is a rather unusual situation. Considering the entire set of experimental results listed above suggests that the mobility of a proton in melanin is associated with a mechanism that does not depend on increasing the water content. Water is needed only to increase the concentration of mobile protons, but not necessarily to increase their mobility. The key factor determining a proton’s mobility in a medium is the distance between local states, i.e., groups capable of taking charge. The standard water-independent distances for proton hopping/tunnelling in melanin are given by the distance of semiquinone radical migration in the course of a single redox reaction within the polymer chain (Figure 9) and by the fixed distances between carboxylic acid groups, which are covalently bound within a melanin particle at the distinct locations and cannot migrate (Figure 10).

To finalize our reasoning, the two-step dissociation of protons in the material from –COOH and from semiquinone may explain the two-step increase of conductivity during hydration observed earlier in melanin [21,23,63]. However, this cannot explain the, although not a large, but statistically significant, drop in conductivity between these two rises. One potential reason could be Zundel cation formation, which hinders proton mobility in solid-state acids [88,89,90]. However, the observation of a Zundel ion spectral feature in a complex system such as melanin is a serious challenge. For example, the band 1724 cm^−1^ is very close to the Zundel cation fingerprint [91], but it is likely a C=O stretching in carboxyl groups in our measurements. Some phenomenology featuring Zundel and hydronium cations was demonstrated in the recently published INS study [62]. However, since it was carried out at 20 K, the observed hydration dynamics of the bands related to aqueous proton cations cannot be extrapolated with confidence to room temperature. In this view, additional far-infrared and terahertz range studies may assist. We also note that another potential entity that can explain the above discrepancies is the recently proposed nitrogen defect structures mentioned above (N–H_2_^+^) [84,85]. The presence of such moieties would undoubtedly affect the CPET mechanism. However, we currently will not speculate further on these entities given the lack of information on their prevalence, chemistry (e.g., pKa), and distribution.

A natural question to ask is how the above assignments compare to computational results. There have been several published studies devoted to the structural calculations of the melanin conformers [92,93,94] using different approaches to quantum chemistry. Several attempts were also made to calculate the normal modes of the solitary melanin monomers [52,59,95]. Unfortunately, none of the studies yielded results that reproduce the experimental data of the polymer system such as for example our synthetic eumelanin material. In the article by Powell et al. [95], the calculations were not compared with experimental results. Unfortunately, this does not conform to the main results and conclusions of our study since it does not regard DHICA monomers. Hyogo et al. [52] compared with experiments, but the authors noted that an additional scale must be used to make their calculations match the experimental results. Probably the most successful in the series from the viewpoint of monomer infrared spectroscopy, Okuda et al. [59] measured the spectra of DHICA and found it was in good agreement with the calculations. However, Okuda et al.’s result did not replicate well the spectra of synthetic or natural melanins with their much more varied types of monomer moieties [48,56,57]. Therefore, despite the success of techniques such as density functional theory (DFT) in calculating basic melanin monomer and oligomer structures, replicating melanin’s IR spectra is still a challenge. The basic root cause is in the algorithm of normal modes’ calculation. DFT-based structure optimization implies seeking the minimum point of the potential energy surface (PES). In its turn, the calculation of normal modes demands information about the PES at the vicinity of the local minimum to calculate its second derivative [96]. Hence, the accuracy of PES calculation in the corresponding area must be high, which is currently not adequate enough. Overall, quantum chemistry *ab initio* calculations of normal modes within melanin are being investigated, though the results have been indifferent. To obtain reliable results in modelling, several key factors will be needed: the calculation of an adequate, representative eumelanin structure; the calculation of said structure when hydrated; the accurate calculation of the IR spectra of eumelanin in the presence of water. Even though we would be naturally interested in confirming theoretical results with experiments, we believe this task should become feasible soon, but it is not feasible computationally right now. Hence, our focus is on a more phenomenological approach in this work. Indeed, this systematic study should prove useful to computational specialists to model the IR spectrum of eumelanin.

In our opinion, the two most clear results of this study are the chemical confirmation of the model of the hydration-dependent comproportionation reaction in the solid state and the demonstration of the importance of the presence of carboxyl groups, which are the main sources of protons at low water concentrations in melanin. This has additional support from previous conductivity work on synthetic melanin synthesized under high oxygen pressures, where the authors found that oxygenated melanins with higher COOH content were more conductive than standard synthesized melanins [85]. The above results tend to indicate that the properties of DHICA within melanin are of great interest since it contributes to the carboxylic groups to the polymer. Clearly, a DHICA-rich melanin should increase the concentration of protons relative to a DHI-rich melanin, leading to higher conductivity, the latter property being of great interest to applied melanin research in recent years. This work does not overturn the current comproportionation model for proton conductivity, but instead strengthens it, albeit with a slight modification. Indeed, the contribution to the conductivity from the protons of the semiquinone is well established and is significant.

Both discussed proton transfer mechanisms, via semiquinone migration and through the hopping between spatially fixed carboxylic groups, are testable. For example, varying the concentration of both carboxyl groups and semiquinones is achievable synthetically. The main experimental problem, though, would be an accurate assessment of mobility or an accurate assessment of the concentration of charge carriers. As such, effort should become a priority to obtain reliable results in the solid state for these two latter key parameters while not relying on expensive and limited techniques such as muSR.

## 4. Conclusions

In conclusion, we presented a hydration-dependent MIR study on eumelanin. We identified the spectral features associated with the presence of specific “one and a half” C𝌁O bonds, typical for *o*-semiquinones. We also revealed the signatures of semiquinone monomers with internal hydrogen bonds and of carboxylic groups. As far as we know, this is the first demonstration of a chemical signature for the hydration-dependent comproportionation reaction in the solid state of eumelanin. In contrast to semiquinones, carboxylic groups dissociate at the beginning of melanin hydration. We propose a modification to the common hydration-dependent conductivity mechanism. We suggest that the conductivity at low hydration is dominated by carboxylic acid protons, whereas higher hydration levels see semiquinone protons manifesting.

## Figures and Tables

**Figure 1 polymers-13-04403-f001:**
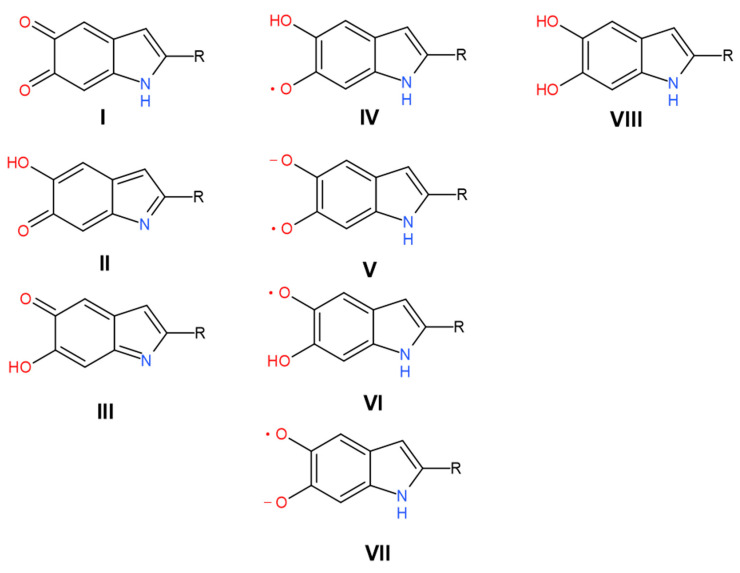
Redox states and tautomeric forms of monomers within the eumelanin chain. R is H (for DHI) and COOH (for DHICA). The left column includes fully oxidized forms: **I**—quinone, **II**—quinone methide, **III**—quinone imine. The central column contains semi-reduced radical forms: **IV**, **VI**—protonated semiquinones, **V**, **VII**—deprotonated radicals. The right column contains a single fully reduced form: **VIII**—hydroquinone.

**Figure 2 polymers-13-04403-f002:**
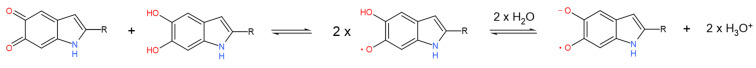
The comproportionation reaction, where the oxidized form (quinone **I**, or quinone methide **II**, or quinone imine **III**) and the reduced form (quinol **VII**) of the moieties leads to the formation of the intermediate oxidation form, the radical semiquinone (**IV** or **VI**). In the solid state, hydration leads to deprotonation of semiquinone to form the semiquinone anion (**V** or **VII**) and mobile proton species, traditionally signified on schemes by hydronium cations H_3_O^+^.

**Figure 3 polymers-13-04403-f003:**
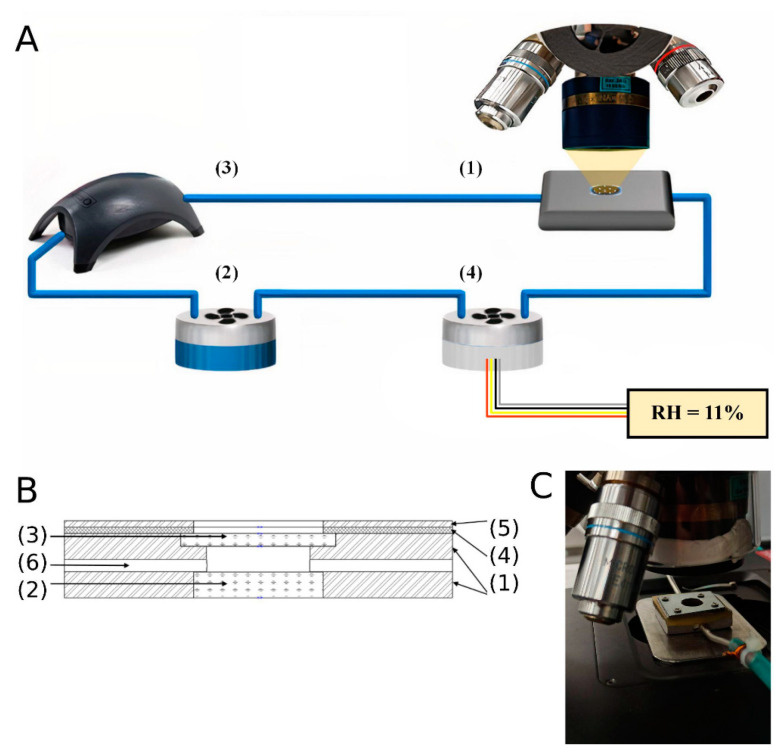
Scheme of the experimental setup. The general scheme of our setup is shown in inset (**A**). Included is an optical hygrocell with a controllable humidity level (1); a container with a saturated salt solution (2) equipped with a fan to ensure a steady spatial distribution of the moisture; a recirculating pump (3); a humidity sensor (4) installed in an empty container right before the inlet of the optical cell. The inset (**B**) is the scheme of the optical hygrocell: the body (1) of the cell was made of low-carbon structural steel; the window (2) was made of CaF_2_. It was glued to the body with epoxy resin. The examined samples were sprayed on the bottom of the replaceable window (3), also made of CaF_2_. The window (3) was sealed with a rubber gasket (4) and fixed by a steel cover (5). Moisturized air was supplied into the hygrocell through the central 1 mm-wide channel (6). (**C**) Photo of the optical hygrocell directly under the microscope objective.

**Figure 4 polymers-13-04403-f004:**
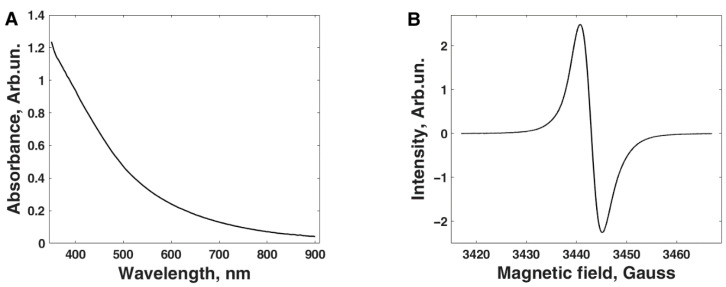
(**A**) An example UV-Vis absorbance spectrum obtained for the melanin sample. The curve shows a simple decaying exponential as expected for the material. (**B**) An example CW-EPR X-band spectrum obtained for the sample.

**Figure 5 polymers-13-04403-f005:**
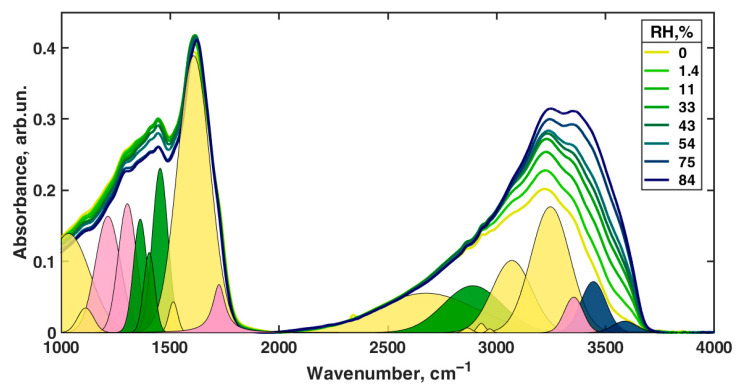
The solid lines are the measured infrared spectra of the synthetic melanin thin film at 25 °C for various hydration levels from 0% RH to 84% RH (see the legend). The colored peak areas result from the 0% RH spectra deconvolution into an appropriate set of well-known peaks. The pink areas include lines associated with the carbonyl vibrations; the two blue peaks are assigned to water-connected vibrations; green areas include vibrations of semiquinone and the remainder; yellow peaks refer to the excitations of the melanin. See the main text for the assignments.

**Figure 6 polymers-13-04403-f006:**
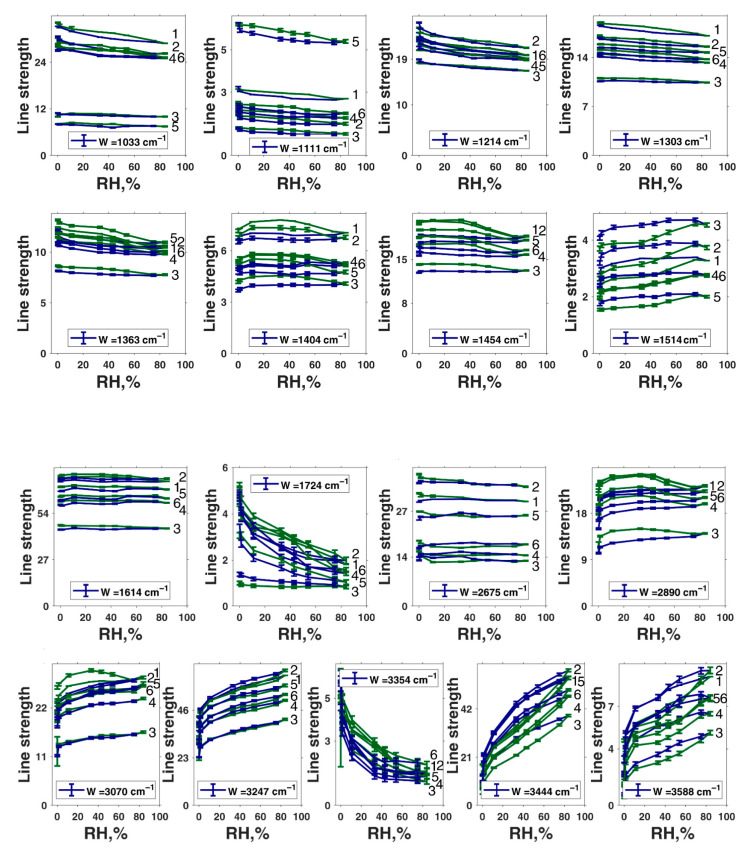
The observed hydration level evolution of the line strengths of all modes in the fitting model. In figure captions, W refers to the central wavenumber of the mode in cm^−1^. The error bars were estimated from the L-M LSDM algorithm. The six lines on each inset numbered on the right refer to the six measured positions on the film. Each line consists of the green and blue parts: the green part corresponds to the RH change from 0% to 84%; the blue part corresponds to the RH change from 84% to 0%.

**Figure 7 polymers-13-04403-f007:**
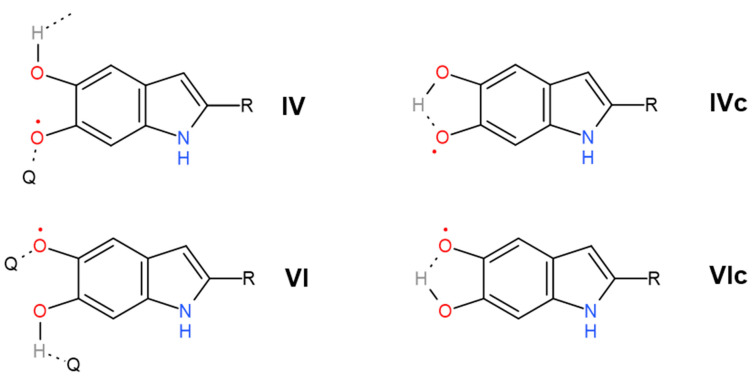
Introduction of the cycled forms of semiquinones, **IVc** and **VIc**. The non-cyclic forms **IV** and **VI** are taken from Figure 1 for comparison. The intramolecular hydrogen bond leads to the hindering of the corresponding O–H and C𝌁O vibrations. Within the framework of our interpretation, the cyclic form of semiquinone is mainly synthesized at the interface since, in this region, the concentration of other melanin units (Q) with which hydrogen bonds’ formation may happen is reduced. Following [83], the activation barrier of the transformation between **IVc** and **VIc** (or reverse chemical reaction) can be crossed at room temperature. Therefore, one of these forms can be redundant.

**Figure 8 polymers-13-04403-f008:**
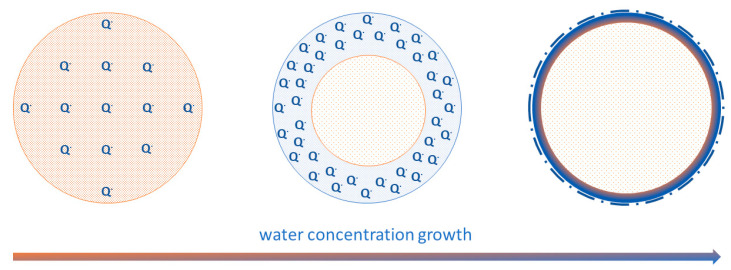
Redistribution of semiquinone monomers in melanin particles from the bulk to their interface caused by water concentration growth. A section of a melanin particle is shown schematically. Semiquinone monomers are shown in blue. The oxidized monomer matrix is shown in light brown.

**Figure 9 polymers-13-04403-f009:**
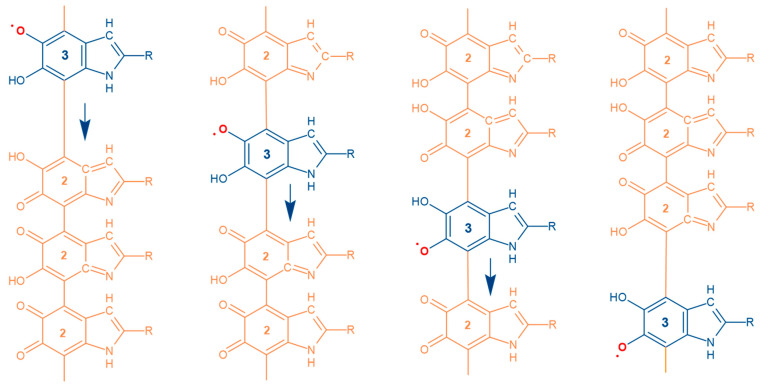
Semiquinone migration is shown as a blue monomer within orange oxidized monomers. The numbers correspond to the number of hydrogens within an indolequinone moiety. Blue arrows indicate the direction of movement of the semiquinone along the chain. The migration of semiquinone along the polyconjugated monomer chain can be represented as a concerted proton–electron transfer (CPET) [87]. In this case, the mobility of protons is strictly connected to the mobility of electron radicals within the chain.

**Figure 10 polymers-13-04403-f010:**
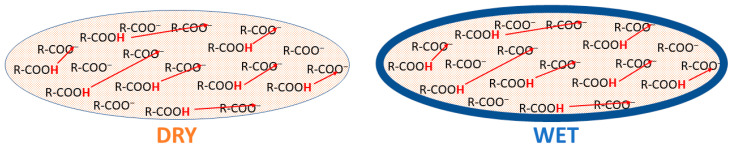
Carboxylic groups are strictly fixed inside the melanin particles and cannot migrate when the humidity changes. If the distances between carboxyls do not change, then the mobility of protons, determined by these distances between local states, should remain almost stable.

**Table 1 polymers-13-04403-t001:** A list of the saturated salt solutions and their corresponding relative humidities.

Salt	LiCl	MgCl_2_	K_2_CO_3_	Na_2_Cr_2_O_7_	NaCl	KCl
RH at 25 °C, %	11	33	43	54	75	84

**Table 2 polymers-13-04403-t002:** The atomic composition (atomic concentration %) and atomic ratios determined from pressed powder pellets of the synthetic melanin sample. For comparison, the expected ratios for the monomer building blocks DHI and DHICA are shown. We note that hydrogen is not included since XPS is unable to extract hydrogen content from samples.

Sample	C (at%)	O (at%)	N (at%)	C/N	O/N	C/O
DHI—Expected	72.7	18.2	9.1	8	2	4
DHICA—Expected	64.3	29.6	7.1	9	4	2.2
Sample	69.5 ± 0.3	21.2 ± 0.4	9.2 ± 0.2	7.6	2.3	3.3

**Table 3 polymers-13-04403-t003:** The list of the observed excitations with a peak position in the fitting model, the corresponding position of the peaks in the second-derivative graph, the maximum and minimum value of α when the peak in SD is still visible, and the modes’ assignments.

Peak Position in Fit, cm^−1^	Second-Derivative Peak Position, cm^−1^	α_max_	α_min_	Assignment of Roldán et al. [56]	Assignment of Centeno and Shamir [48]	Assignment of Bridelli et al. [61]	Assignment of Perna et al. [57]	Other Assignments	Assignment in the Current Work
1033	1031	151.6	2.7	δ(CH) + δ(NH) + ν(CO)	CH in-plane/CH out-of-plane deformation	Aromatic C–H bending	CH in-plane deformation	C–O in catechol, quinone-imine, and carboxylate [76]	δ(CH) + δ(NH) + ν(C–O)
1111	1102	272.5	2		CH in-plane deformation	Aromatic C–H bending	O–H, C–H, N–H deformation		δ(CH) + δ(NH) + ν(C–O)
---	1165	110.8	0.2	δ(OH) + δ(CH) + δ(NH)	Pyrrole NH in-plane deformation/ring breathing	Aromatic C–H bending	C–H in-plane deformation		δ(OH) + δ(CH) + δ(NH)
1214	1215	99.1	0.1	δ(OH) + δ(CH)	CO stretching/OH in-plane deformation in COOH	Carboxylic C–O stretching or OH bending	C–H in-plane deformation	C^-^ stretching and C–O–H asymmetrical of COOH [15]	δ(OH) + δ(CH) + ν(C(O)–OH in carboxyls) + ν(C–OH in conjugated cycles)
1303	1267	0.5	0.1	ν(CO) + δ(CH) + δ(ring)	CH in-plane deformation	Carboxylic C–O stretching or OH bending	C–C and C–N stretching in pyrrole, C–OH stretching in phenolic group, amide III	Phenolic moieties [77]	ν(CO) + δ(CH) + δring
	1289	400	0.1	ν(CO) + δ(NH) + δ(CH)		Carboxylic C–O stretching or OH bending			Carboxylic C–O stretching or OH bending
	1318	0.3	0.1	Amide III proteins		Carboxylic C–O stretching or OH bending	1310–C–OH stretching in COOH, C–N stretching in pyrrole, C–OH stretching, and O–H deformation combination in phenolic groups		Carboxylic C–O stretching or OH bending
1363	1345	10.1	0.1	ν(CN) + δ(OH) + ν(ring)	Indole ring vibration/CN stretching				ν(CN) + δ(OH) + ν(ring)
	1380	16	0.1	δ(CH2), ν(CC) polysaccharides, ν(CO), δ(CH), δ(CN), δ(NH) proteins	Pyrrole ring stretching		C–N stretching, indole ring vibration		Cyclic semiquinone C𝌁O stretching or C–N stretching, indole ring vibration
1404	1406	1.5	0.1	δ(OH) + ν(ring) [56]	Pyrrole ring stretching	Carboxylate ion symmetrical stretching	C–O symmetric stretching in COOH	phenolic C–O–H bending [15]	Cyclic semiquinone C𝌁O stretching
1454	1443	167	0.1	ν(ring) + δ(CN) + δ(OH)	C=C aromatic ring vibration				Semiquinone C𝌁O stretching
	1468	83.7	0.1	ν(ring) + δ(CN) + δ(OH)	Pyrrole ring stretching vibration		Indole ring vibration, C–C in-plane vibration in pyrrole	lower wavenumber C–C of aromatic C–C moieties [77]	Semiquinone C𝌁O stretching
1514	1518	22.5	0.2	ν(ring) + δ(NH) + δ(CH)			C–H deformation mixed modes, amide II		ν(ring) + δ(NH) + δ(CH) OR semiquinone anion C𝌁O stretching
1614	1592	4.3	0.1	ν(ring) + ν(C=O)	C=C aromatic/pyrrole ring stretching		Indole ring vibration	ionized carboxylic acid (COO–) [53]; aromatic C=C stretching and COO– asymmetrical stretching [15]	ν(ring) + ν(C=O);
	1629	400	0	ν(ring) + δ(CH)		OH bending (H_2_O)	Indole ring vibration	C=C and C=N bending modes and C=O stretching mode from noncarboxylic acid moieties [53]; C=O in catechol, quinone-imine, and carboxylate [76]	ν(ring) + ν(C=O) + OH bending (H_2_O)
	1653	0.3	0.1	ν(C=O)			C=O stretching in quinone		ν(C=C) + ν(C=O)
	1671	1.6	0.1				C=O stretching in quinone		ν(C=C) + ν(C=O)
	1684	0.1	0.1				C=O COOH stretching	C=O stretching mode in –COOH [53]	ν(C=C) + ν(C=O)
	1700	0.1	0.1				C=O COOH stretching	C=O of COOH stretching [15]	ν(C=O)
1724	1717	16.9	0.1				C–O asymmetric stretching in COOH		C=O COOH stretching
	1730	87.5	0.1	ν(C=O) lipids, polysaccharides		C=O COOH stretching	C=O COOH stretching		C=O COOH stretching
	1745	0.1	0.1				C=O COOH stretching		C=O COOH stretching
	1771	0.4	0.1				C=O COOH stretching		C=O COOH stretching
	1882	400	25						sample preparation residuals
---	2092	333.3	12.6						sample preparation residuals
---	2224	233.3	0.6						sample preparation residuals
---	2474	275	38.7					–OH and –NH stretching modes of the carboxylic acid (C–O and –COOH); phenolic (C–O/carboxyl OH) and aromatic amino functions in the indolic and pyrrolic systems present in DHI and DHICA- derivatives [53]	
2675	2598	266.6	11			Carboxylic H-bonded OH stretching		Carboxylic H-bonded OH stretching [15]	Enol H-bonded OH stretching (oxidized monomers II, III on Figure 1)
---	2756	366.6	8.3			Carboxylic H-bonded OH stretching			Carboxylic H-bonded OH stretching
2866	2858	116.6	0.1			Aliphatic C–H stretching			Aliphatic C–H stretching
2890	2878	0.1	0.1	ν_sym_(CH3) lipids	ν(CH)				Cyclic semiquinone O–H stretching (intramolecular H-bond)
	2902	0.9	0.1		ν(CH)				ν(CH)
2929	2930	400	0.1	ν_assym_(CH_2_) lipids	ν(CH)	Aliphatic C–H stretching			Aliphatic C–H stretching
2968	2965	4.3	0.1	ν_assym_(CH_3_) lipids, cholesterol, proteins					Aliphatic C–H stretching
	3018	1	0.3						Aliphatic C–H stretching
3070	3069	400	0.1	ν(=CH)	ν(CH)				Aromatic C–H stretching
3247	3214	400	0.1			N–H–NH_3_^+^ stretching		Water-connected *ν*(OH) [50]	ν(NH) in aromatic system
3354	3361	400	0.1	ν(NH)	ν(NH)			N–H stretching and OH–H-bonded stretching [15]	ν(C(O)O–H) hydrogen bonded
3444	3462	400	0.1			N–H–NH_2_ symmetrical and asymmetrical stretching OR OH–H-bonded stretching		Water-connected *ν*(OH) [50]	OH H-bonded stretching in water
3588	3610	400	0.1					–OH and –NH stretching modes of the carboxylic acid (C–O and –COOH); phenolic (C–O/carboxyl OH) and aromatic amino functions in the indolic and pyrrolic systems present in DHI and DHIC derivatives [53]; water-connected *ν*(OH) [50]	OH stretching in water

## Data Availability

The data that support the findings of this study are available from the corresponding author, K.M., upon reasonable request.

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
