# Peer review of "Water-Activated Semiquinone Formation and Carboxylic Acid Dissociation in Melanin Revealed by Infrared Spectroscopy"

_polymers, 2021, doi:10.3390/polym13244403_

Round 1

Reviewer 1 Report

The work has a comprehensive FT-IR analysis of synthetic melanin at controlled humidity levels. On page 6, the authors mention: "The integral Kratos...", please include a reference.  English must be revised, for example on page 9: " Coupled with more sensitive nature of the SD plot one can extract the peaks, making it profitable", a phrase hard to understand; on page 10: ".. overparameterization". On page 23, fig. 6 shows that many lines have a high dispersion in value depending on the position of the measurement, see for example A or M. In some cases, the line strength varies more than 100%. The authors did not discuss this effect along the interpretation of the data, which should be included. Specially since as already mentioned the heterogeneity is affecting some lines more than others. On page 26, the authors mention active synthesis of semiquinones, probably they ment formation. In a phrase further they associate the water content to uncharged protonated forms, this phrase is confusing. On page 27, with regard to the band 3247 cm-1, did the authors consider the presence of N-defects (see DOI: 10.1039/d1ma00446h)? These defects could play an important role in hydrogen migration, see Fig. 9. On page 32, the authors discuss the role of DHICA on conductivity, which has been studied recently in DOI: 10.1039/d1tc01440d. 

Author Response

The work has a comprehensive FT-IR analysis of synthetic melanin at controlled humidity levels.

ANSWER. We thank the reviewer for their assessment and time in reviewing the manuscript with recommendations to improve the text.

On page 6, the authors mention: "The integral Kratos...", please include a reference.  

ANSWER. We thank the reviewer for this comment. As a result, we have added the reference Baer et al, 2020, “XPS guide: Charge neutralisation and binding energy referencing for insulating samples”, DOI: 10.1116/6.0000057. We have also changed the phrase “The integral Kratos charge neutralizer was used as an electron source to eliminate differential charging”  into (page 8 line 17) “To eliminate differential charging [70], the charge neutraliser integral to the Kratos spectrometer was used as an electron source.

English must be revised, for example on page 9: " Coupled with more sensitive nature of the SD plot one can extract the peaks, making it profitable", a phrase hard to understand; on page 10: ".. overparameterization". 

ANSWER. We thank the reviewer for this comment. The phrase "Coupled with more sensitive nature of the SD plot one can extract the peaks, making it profitable" was deleted, and the word “overparameterization” was changed intomutual dependencies between fitting parameters”.

On page 23, fig. 6 shows that many lines have a high dispersion in value depending on the position of the measurement, see for example A or M. In some cases, the line strength varies more than 100%. The authors did not discuss this effect along the interpretation of the data, which should be included. Specially since as already mentioned the heterogeneity is affecting some lines more than others. 

ANSWER. We thank the reviewer for this comment. This is an important note, and we added a discussion in the text below figure 6 in the following way (page 24 line 6):

"We first should mention that the line's strength dependencies level varies from one measured position to another. And for some modes (for example 1033 cm-1, 1111 cm-1, 2675 cm-1) those changes are larger than 100%. We can explain it by stating that our film is inhomogeneous not only in thickness but also in chemical composition. However, if we normalise each graph to its value at 84% (ESI Fig. 4), we could see that the shapes of the dependencies are consistent between measured positions on the film. It means that local variations of the chemical composition of our film do not affect the chemical processes occurring in our sample on a qualitative level."

In addition, we wish to emphasise again that our sample has various chemical compositions from one measured point on the film to another. This chemical heterogeneity clearly manifests itself in figure 6, but this is exactly as expected and what we want to show. In most cases, we based our interpretation on the normalised results (ESI fig. 4 and 5). Still, in further analyses, we found it possible to consider some details from the not normalised graphs. Therefore, for ease and in light of our new addition,  we will leave figure 6 with unnormalised graphs in the main text. 

On page 26, the authors mention active synthesis of semiquinones, probably they ment formation. 

ANSWER. We thank the reviewer for this comment. We changed the word “synthesis” for “formation”.

In a phrase further they associate the water content to uncharged protonated forms, this phrase is confusing. 

ANSWER. We thank the reviewer for this comment. We have changed the corresponding phrases in order to make the meaning clearer. It now reads (page 26 line 20):  

This is supported by the rise of the curve for the bands at 1404 and 2890 cm-1 in the range 0-33% RH. A further increase of water concentration (RH values above 33%) should continue to catalyse the comproportionation reaction. However, it also leads to partial deprotonation of the neutral semiquinone, both with intra- and intermolecular hydrogen bonds, and to the formation of the semiquinone anion (Fig.1, VII). Depletion of the protonated semiquinone forms decreases the line strength of corresponding bands (1404, 1454, 1468, 2890 cm-1) for RH values above 33%.

On page 27, with regard to the band 3247 cm-1, did the authors consider the presence of N-defects (see DOI: 10.1039/d1ma00446h)? These defects could play an important role in hydrogen migration, see Fig. 9. 

ANSWER. We thank the reviewer for this comment. We added the following considerations to the Results and Discussion section. The first section was added to address the first comment (page 27 line 23): 

Another potential option is a recently proposed radical moiety within melanin, a nitrogen defect, which would yield a protonated amine (N-H2+) [84], [85]. At this point in time, there is no further information on these structures except as suggested by electron paramagnetic resonance. We also infer that if having to choose between N-H and N-H2+, the protonated unit would likely be a significantly smaller population vis-à-vis the amine and as such, we are inclined to ascribe the signal to the amine.” 

We have added the following section to address the second part of the comment (page 32 line 20): 

We also note that another potential entity that can explain the above discrepancies is the recently proposed nitrogen defect structures mentioned above (N-H2+) [84], [85]. The presence of such moieties would undoubtedly affect the CPET mechanism. However, we currently will not speculate further on these entities given the lack of information on their prevalence, chemistry (e.g. pKa) and distribution.

On page 32, the authors discuss the role of DHICA on conductivity, which has been studied recently in DOI: 10.1039/d1tc01440d. 

ANSWER. We thank the reviewer for this comment. We modified and added the following text to the Results and Discussion section (page 33 line 30).

“This has additional support from previous conductivity work on synthetic melanin synthesized under high oxygen pressures, where the authors found that oxygenated melanins with higher COOH content were more conductive than standard synthesised melanins [85].”

Reviewer 2 Report

In this study the Authors have presented the results of the FT-IR analysis of eumelanin at different RH, conducted in order to explain the reaction mechanism. The submitted manuscript is scientifically sound and clear. However, I think it would benefit from revision made after reading the comments presented below.

The continuous line numbering is missing. This would really facilitate the review process.

Page 5, “As previously demonstrated”-what do you mean by that?

Page 9, have you used only pure Gaussian or pure Lorentzian for the peaks or the combination of those two (for one peak)?

Table 2, the differences between the measured and expected elemental composition should be explained.

Table 3, a dot should be used as a decimal separator

Table 3, why some of the peaks were left unassigned? I.e. 1771, 1882?

Figure 6, I don’t understand the reason for those letters (A-Q)?

Page 5, it should be X-ray.

Table 2, what about the Hydrogen content?

In the manuscript the Authors don’t mention the computational results on the melanin. A lot of them have been shown to provide the answers to the structural problems. Since it is possible to simulate the FT-IR spectra using i.e. DFT calculations, wouldn’t it be reasonable and feasible to confirm your findings by applying properly performed quantum mechanical simulations?

Author Response

In this study the Authors have presented the results of the FT-IR analysis of eumelanin at different RH, conducted in order to explain the reaction mechanism. The submitted manuscript is scientifically sound and clear. However, I think it would benefit from revision made after reading the comments presented below.

ANSWER. We thank the reviewer for their assessment, time, and careful evaluation of the manuscript.

The continuous line numbering is missing. This would really facilitate the review process.

ANSWER. We thank the reviewer for this comment. We added continuous line numbering.

Page 5, “As previously demonstrated”-what do you mean by that?

ANSWER. We thank the reviewer for this comment. The phrase was deleted.

Page 9, have you used only pure Gaussian or pure Lorentzian for the peaks or the combination of those two (for one peak)?

ANSWER. We thank the reviewer for this comment. We used Gaussian line shapes for all bands except one with the 1724 cm-1 peak position, as stated in the manuscript were we write (page 9 line 9):

We found that a set of Gaussian lines fits well with the experimental data except for the mode with 1724 cm-1 peak position.

We noted a deficiency in the text around this point and corrected it as follows (page 9 line 3):

 “The procedure involves collecting the peak positions in the SD graph for spectra, measured at all six spatial positions on the sample and all humidity levels. We then used the SD peak positions as an initial guess for the peak positions in the fitting procedure for the first measured position on the film and for RH = 0% for the Gaussian lines basis set.

Table 2, the differences between the measured and expected elemental composition should be explained.

ANSWER. We thank the reviewer for this comment. We rewrote the results and discussion section devoted to XPS characterisation of melanin and added the following (page 10 line 23):

Melanin is an oligomeric mixture primarily composed of DHI and DHICA moieties and their assorted oxidative states [11], [72], [73]. Consequently, it is expected that melanin samples will have an overall atomic ratio profile that falls in between that expected for DHI and DHICA (see Table 2, “Expected”). The XPS data indicates that the sample is compatible with that of synthetic melanin (Table 2, see “Sample”). The sample is closer to the ideal DHI values, indicating the dominant ion of DHI moieties. This is expected for the synthesis procedure employed here [66] since it is known that there is a loss of COOH units as well as the formation of a small amount of smaller ring units due to degradation [74]. We also note that elemental surface scans of melanins are representative of the bulk, as previously demonstrated [75]. Overall, the characterisations above show that our sample is a synthetic sample of eumelanin.

Table 3, a dot should be used as a decimal separator

ANSWER. We thank the reviewer for this comment. Dots were replaced with decimal separators. 

Table 3, why some of the peaks were left unassigned? I.e. 1771, 1882?

ANSWER. We thank the reviewer for this comment. We revised the table and added assignment from the literature source for the 1771 cm-1 peak. For the modes with central frequencies 1882 cm-1, 2092 cm-1, and 2224 cm-1 we added the following comment (page 22 line 1): 

“We did not find appropriate assignments for modes with central frequencies 1882, 2092, and 2224 cm-1. However, in [56], one may observe several broad and weak peaks in the region 1800 - 2300 cm-1. More important is that the number and position of those peaks enormously vary from one sample to another. That is why we conclude these peaks are due to some residuals of synthesis.” 

Figure 6, I don’t understand the reason for those letters (A-Q)?

ANSWER. We thank the reviewer for this comment. We agree that this indexation of insets is redundant and deleted it.

Page 5, it should be X-ray.

ANSWER. We thank the reviewer for this comment. We have corrected the spelling.

Table 2, what about the Hydrogen content?

ANSWER. We thank the reviewer for this comment. We respectfully point out that XPS is incapable of hydrogen content extraction. In order to prevent a similar question from the reader, we have inserted the phrase:

 “We note hydrogen is not included since XPS is unable to extract hydrogen content from samples.” 

into the caption of table 2.

In the manuscript the Authors don’t mention the computational results on the melanin. A lot of them have been shown to provide the answers to the structural problems. Since it is possible to simulate the FT-IR spectra using i.e. DFT calculations, wouldn’t it be reasonable and feasible to confirm your findings by applying properly performed quantum mechanical simulations?

ANSWER. We thank the reviewer for this comment. To highlight the problems of the computational chemistry of melanin, we have added the following text to the discussion (page 32 line 25):

A natural question to ask is how the above assignments compare to computational results. There have been several published studies devoted to the structural calculations of the melanin conformers [92]–[94] using different approaches to quantum chemistry. Several attempts were also made to calculate the normal modes of the solitary melanin monomers [52], [59], [95]. Unfortunately, none of the studies yielded results that reproduce the experimental data of the polymer system such as for example our synthetic eumelanin material. In the article by Powell et al. [95] the calculations were not compared with experimental results. Unfortunately, it does not conform to the main results and conclusions of our study since it doesn’t regard DHICA monomers. Hyogo et al. [52] compare with experiments, but the authors noted that an additional scale must be used to make their calculations match the experimental results. Probably the most successful in the series from the viewpoint of monomer infrared spectroscopy, Okuda et al. [59] measured the spectra of DHICA and found it was in good agreement with the calculations. However, the Okuda et al. result did not replicate well the spectra of synthetic or natural melanins with their much more varied types of monomer moieties [48], [56], [57]. Therefore, despite the success of techniques such as density functional theory (DFT) in calculating basic melanin monomer and oligomer structures, replicating melanin’s IR spectra is still a challenge. The basic root cause is in the algorithm of normal modes calculation. DFT-based structure optimisation implies seeking the minimum point of the potential energy surface (PES). In its turn, the calculation of normal modes demands information about the PES at the vicinity of the local minimum to calculate its second derivative [96]. Hence, the accuracy of PES calculation in the corresponding area must be high, which is currently not adequate enough. Overall,  quantum chemistry ab initio calculations of normal modes within melanin are being investigated, though results have been indifferent. To obtain reliable results in modelling, several key factors will be needed: calculation of an adequate, representative eumelanin structure; calculation of said structure when hydrated; accurate calculation of the IR spectra of eumelanin in the presence of water. Even though we would be naturally interested in confirming theoretical results with experiments, we believe this task should become feasible soon, but it is not feasible computationally right now. Hence, our focus is on a more phenomenological approach in this work. Indeed, this systematic study should prove useful to computational specialists to model the IR spectrum eumelanin.

Round 2

Reviewer 1 Report

The authors did answer and revise the work accordingly to my previous report. The work is an important contribution to the researchers working in the field. The work has a comprehensive FT-IR analysis of synthetic melanin at controlled humidity levels.

Reviewer 2 Report

The Authors have revised and improved their manuscript, therefore this version can be accepted as it is.